# COMMAT: DATASETS AND BENCHMARKS FROM COMPLEX MATERIALS FOR GRAPH MACHINE LEARNING

## ABSTRACT

Recent research has demonstrated the efficacy of graph learning over a wide spectrum of materials, including molecular graphs, crystals, mechanical metamaterials, and strongly disordered systems. In this work, we draw attention to the broad class of *complex materials*, which combine order and disorder, and fall outside the above categories, yet have shown superior properties throughout the materials science literature. We present a Complex Material Benchmark (ComMat), including three graph datasets of complex materials from experimental and computational research studies, unifying distinctly developed data-to-graph pipelines under a standardized graph-based representation. In particular, we provide the first publicly available 3D graph dataset of a nanoscale network derived from 3D tomography. We then quantitatively show that these graphs are fundamentally different from existing materials datasets. We design various predictive tasks to advance machine learning (ML) methods, including experimentally measured properties, simulated mechanical response, and structural awareness. Extensive benchmark experiments are conducted over popular graph learning models, revealing their limitations and the need for further development in handling complex materials. ComMat is openly released to accelerate ML research and innovation in complex material design.

## 1 INTRODUCTION

The discovery of high-performance materials is often the bottleneck in the development of tools in science and engineering applications. Such materials must often combine contradictory properties. Examples include coatings that must simultaneously be electrically conductive, flexible, and transparent, for applications in flexible electronics; or structural batteries that must be simultaneously load-bearing, ionically conductive, but electrically insulating. With advancements in synthesis and manufacturing technologies, our ability to create these materials increasingly "outpaces our ability to test them" (Sarkisov & Kim (2015)). This has driven the development of many computational tools for studying structure–property relationships in modern materials, including inverse design methods that tailor structures to achieve desired properties. However, even with these advances, progress remains constrained by the high cost of simulating systems at realistic scales and with diverse structural details. These limitations have motivated the development of machine learning (ML) for materials science (Choudhary et al. (2022)).

Graph representations offer a flexible framework for encoding material structures across scales, enabling deeper exploration of their structure–property relationships. Data from experiments (e.g., microscopy images) and simulations (e.g., molecular dynamics trajectories) can be readily mapped to graphs, enabling the application of network science and graph learning methods to uncover deeper physical insights. However, most studies are focused on crystalline materials (Yan et al. (2022)) and molecular graphs (Gilmer et al. (2017)). In each case, the material can be well defined by a relatively small amount of information, either by using unit-cells for crystals or molecular graphs for small molecules ($\sim$100 nodes). This greatly simplifies graph representations and subsequent ML tasks.

Our first claim is that there is a very broad category of practically relevant materials that do not fall into the above categories and are not currently investigated with ML. Specifically, we are concerned with *complex materials*. In this context, *complex* refers to the combined presence of *order and disorder* in a material's structure (Mao & Kotov (2024)). This structural characteristic has been shown to

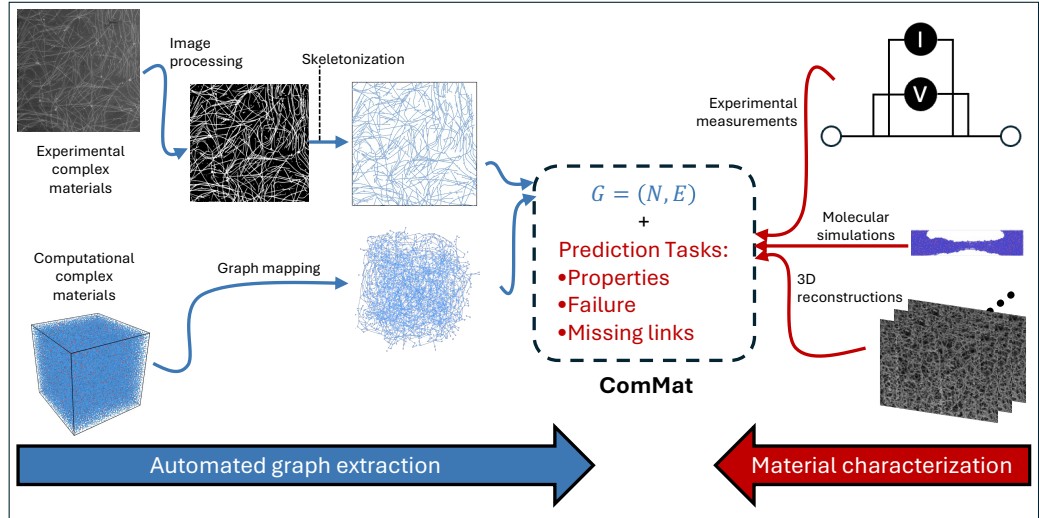

Figure 1: Left: Graph representations of complex materials derived from multi-modalities - experimental microscopy of a 2D percolating network of nanowires processed using *StructuralGT* (top) and molecular dynamics trajectories (bottom). Right: Information from complex material characterization, from top to bottom, experimental measurements of properties, molecular simulations of fracture, 3D imaging. Middle: Combination of complex material graphs and ground truth information to form ComMat samples. Each ComMat sample involves predicting complex material behavior from the associated graph, such as experimentally measured resistances, locations of failing polymer links in molecular simulations, or 3D network structure as imaged by microscopy. As a further example, we include an excerpt of a 3D graph in Appendix A.

impart many superior material properties of great importance in high-performance applications, including improved transport properties (Smart et al. (2007)), fracture toughness (Fulco et al. (2025)), anisotropy (Wu et al. (2024)), and optical activity (Kuznetsova et al. (2025)). Such materials are also synthetically more accessible, which makes them practically attractive for low-cost manufacturing and future technologies. Simultaneously, complex materials are a predominant form in Nature and so are also of fundamental interest (Jiang et al. (2020)).

However, the opportunities raised by the prevalence and utility of material complexity simultaneously raises challenges with regard to their ML-driven study. Specifically, the most efficient representations of complex materials remain challenging due to the combined order and disorder inherent in their structure (Mao & Kotov (2024)). The multi-scale nature of complex materials and the absence of a generalizable theory make their structure–property relationships difficult to explore (Ortiz-Tavárez et al. (2025)). The commonly used unit cell representations do not apply due to the lack of periodic order in complex materials. Given that complex materials are harder to generate from toy models, there is also a stronger reliance on generating them from either costly experiments or computationally expensive simulations. The resulting data scarcity motivates both centralizing the data that is already available, and subsequently developing one-shot learning models for their analysis.

One of the most promising frameworks for complex material study involves graphs, allowing for a universal representation while flexibly encoding structural details at different levels through adjustable graph properties. Recent advancements in graph-based methods have enabled quantitative study of complex materials for both experimental and computational researchers. Firstly, the development of *StructuralGT* (Vecchio et al. (2021)) - an image informed package for generating and analyzing graphs - has allowed experimental researchers to rapidly extract graphs from images. This process is depicted in the top-left quadrant of Figure 1. Meanwhile, the increasing availability of computing resources has pushed researchers to simulate increasingly complex systems. With such increasing complexity, some researchers have found benefits of graph-based representations for detecting phase transitions (Yang et al. (2024)), probing structural changes (Choi & Cho (2016)), and predicting properties (Zhang & Riggleman (2024)). In these cases, simulation trajectories are con-

verted to graphs directly from the system's bond information, as depicted in the bottom-left quadrant of Figure 1. We argue that a unified graph-based representation for these complex material - in computational and experimental structures - allows us to apply findings in the broadest possible manner. For example, the ability of Geodesic Edge Betweenness Centrality to predict failure locations in various materials has been confirmed by separate research groups in computational (Mangal et al. (2023); Zhang & Riggleman (2024)) and experimental studies (E. Berthier (2019)).

One of the greatest challenges associated with building a complex material ML benchmark is the distinctly high cost associated with collecting complex material data. Graphs from experimental samples must first be *synthesized*, requiring specialized laboratory expertise, equipment, and materials. Then they must be *imaged*, requiring expensive and difficult to operate microscopy equipment. Finally, *image-to-graph extraction* requires novel image processing including binarization, skeltonization, and node/edge detection. Python APIs with ML library compatibilities, such as *StructuralGT*, have only just been released. For graphs derived from molecular simulations, developing a physically reliable simulation pipeline, including critical steps such as force-field parameterization and validation of model assumptions, requires significant time and specialized expertise. Rigorous post-processing is also needed to ensure that the extracted networks are physically meaningful. In both experimental and computational cases, defining a practically relevant task associated with each material requires intimate knowledge of the material properties and industries that find them interesting. Combined with all the above necessitates highly interdisciplinary collaborations, which is an additional challenge.

This further motivates the current cohesive, interdisciplinary, and open-source dataset for complex material ML-driven study. We take raw data from both the computational and experimental pipelines mentioned above, and unify them under the general $G = (N, E)$ representation, whilst maintaining geometric information via node position attributes. The complex materials included in our study are chosen to satisfy two criteria: (1) adherence to our definition of complexity, involving the co-existence of ordered and disordered structural features, and (2) practical significance in real-world applications. Our dataset comprises a nanofiber graph containing more than 330,000 nodes, samples extracted from experimental nanowire images, and molecular simulations of polymer networks. It enables multiple ML tasks aligned with practical material applications, including fracture prediction (edge-level), resistance prediction (graph-level), and link prediction, as depicted in the right half of Figure 1. By evalutaing fundamental structural parameters, we quantitatively show that each of our complex materials is fundamentally distinct from materials reported in previous databases.

Our contributions are as follows. 1) We introduce ComMat, a collection of three graph datasets from complex materials, integrating both 2D and 3D structures from experimental and computational sources. We provide complexity analysis to distinguish ComMat properties from existing materials datasets. To the best of our knowledge, this is the first open-source benchmark of complex materials at this scale. 2) We design a diverse set of predictive tasks, ranging from edge-level to graph-level, to drive ML advancements in materials science. 3) We conduct extensive experiments to benchmark a wide range of mainstream graph learning baselines on our proposed tasks, establishing a robust foundation for future research.

## 2 BACKGROUND & RELATED WORK

### 2.1 BACKGROUND ON ORDER, DISORDER, & COMPLEXITY

When a material has translational order, it is said that its composition is identical at points that are apart by some given separation: $f(\mathbf{r} + \mathbf{R}) = f(\mathbf{r})$, where $f$ returns the vector that defines the content of the material at a particular point, $\mathbf{r}$ is an arbitrary position, and $\mathbf{R}$ is some separation vector. This allows for an efficient representation of the structure because one only needs to define the unit cell and the manner in which it can be used to reconstruct the entire material. Such representations were used by Yan et al. (2022) who represent atomic crystalline unit cells with $\mathbf{A}, \mathbf{P}, \mathbf{L}$, where each of the matrices is a set of vectors identifying the contents, positions and manner of the unit cell repeated to form the bulk material. Subsequent work by Liu et al. (2022) included further geometric features, such as bond angles, while work by Yan et al. (2024) achieves SE(3) invariant and SO(3) equivariant representations. Similar methods have been applied to polymer crystalline materials (Zeng et al. (2018)), while extensions to weakly disordered materials involve training on their strictly ordered

counterparts, followed by extensions that accommodate the minor deviations from perfect order (Chen et al. (2021); Wang et al. (2022); Eremin et al. (2024)).

Just as is the case with the above ordered materials, when materials are strongly disordered, their behavior can be inferred from their local environment (Kim et al. (2023)). As a result, graph learning studies have allowed researchers to differentiate liquids from amorphous solids (Swanson et al. (2020)), predict their long-time evolution (Bapst et al. (2020)), and inversely design their structure for mechanical property improvements (Wang & Zhang (2021)). Local environment representations have also been shown to reproduce mechanical properties (Sak (2025)) and short-range order in high entropy alloys (Sheriff et al. (2024)).

When materials are neither ordered nor disordered, but instead complex, their local behavior is often dependent on structural features beyond just their local environment. This phenomenon has been demonstrated by Smart et al. (2007), who show that particles with high betweenness centrality values experience higher heat flow than those with high nodal degree. Similarly, concentration of stress in granular media and strut lattices is dependent on global connectivity patterns, as captured by betweenness centrality metrics (Kollmer & Daniels (2019); E. Berthier (2019); Reyes-Martinez et al. (2024)).

By examining the definition of betweenness centrality, we may learn the problems with applying popular unit-cell like graph representations of complex materials: Edge betweenness centrality, $EBC_G$, of an edge, $e$, is defined as

$$\text{EBC}_G(e) = \frac{1}{N(N-1)} \sum_{s,t \in N} \frac{\sigma_{st}(e)}{\sigma_{st}} \tag{1}$$

where N is the number of nodes. $\sigma_{st}$ is the number of shortest paths between a given source and target, $s$, $t$, respectively. $\sigma_{st}(e)$ is the number of shortest paths between $s$, $t$ containing $e$. So given that the local betweenness value depends on the entire graph, because of the dependence on $\sigma_{st}$, learning betweenness centrality for complex materials would require inefficiently numerous iterations of the 1-hop neighbor information exchange scheme that is central to graph learning techniques that use the message-passing (MP) paradigm (Gilmer et al. (2017)). This is because each iteration of the MP paradigm only augments the representations of each node with its immediate neighbors. It is this defining feature of local behavior depending on global connectivity that often separates complex materials from their ordered and disordered counterparts and thus motivates our complex material dedicated dataset.

## 2.2 RELATED WORK

In preparing this dataset, we surveyed the current literature on graph learning material datasets. We learnt that all previous studies use the message passing paradigm, which cannot easily learn centrality parameters, which have previously shown to impact complex material properties. We also learnt that many rely on unit-cell representations that are incompatible with our materials. Finally, we learnt that there are no existing benchmarks for complex materials. Below we review some existing datasets and benchmarks to highlight the gap in the literature that our work fills. Further related work is discussed in Appendix D.

**Mechanical Metamaterials** A popular class of materials for graph machine learning is metamaterials, defined as those with interesting properties arising from their precisely arranged geometry (Fang et al. (2022)). Having exploded in recent years due to advances in 3D printing and additive manufacturing, there is increasing demand for their computational generation and analysis. Works targeting metamaterial design often use experiments or finite-element simulations for establishing ground truth target properties and behavior, such as MetaMatBench (Chen et al. (2025a)), which standardizes 5 diverse metamaterial datasets into a unified representation $M = (L, U, y)$. They also provide a robust toolbox with 17 adapted ML models for property prediction and inverse design, alongside a novel evaluation suite featuring 12 metrics and finite-element analysis for physics-aware validation. Zhan et al. (2025) then extend this work to collectively consider all modalities associated with ordered mechanical metamaterial design.

Although mechanical metamaterials are not necessarily ordered, the above works focus on ordered materials, and thus, rely on unit-cell based graph representations. While Grega et al. (2024) include

nodal perturbations in their GNN trained dataset, the importance of tuning disorder beyond the mechanical metamaterial unit cell is highlighted by Fulco et al. (2025), who show that its presence enhances fracture toughness. In other words, identifying the best candidates must involve inclusion of complex materials, relying on graph representations incompatible with the unit-cell approach used in the above works.

**Nanomaterials**  In a similar vein to metamaterials, there are numerous studies of crystalline nano-materials whose assumption of order enables unit-cells for efficient graph-based representations. Instead of finite-element, the ground truth for most ML tasks comes from Density Functional Theory (DFT). ECD Chen et al. (2025b) proposes large-scale datasets for electronic charge densities prediction. It contains 140,646 crystal geometries with medium-precision calculations and a subset of 7,147 geometries with high-precision data. Friis-Jensen et al. (2024) present two novel datasets for nanomaterials: CHILI-3K, a medium-scale dataset of over 6 million nodes focused on mono-metallic oxides, and CHILI-100K, a large-scale dataset of over 183 million nodes derived from experimentally determined crystal structures with broad chemical diversity. They also define several property and structural prediction tasks, benchmarking a wide array of GNN models. Instead of studying bulk crystals, they study small crystallites whose entirety can be represented with a graph.

## 3 COMMAT DATASETS

In Sections 3.1–3.3, we detail three datasets from complex materials with broad real-world applications: polymer networks obtained from 3D molecular dynamics simulations, silver nanowire networks derived from 2D microscopy images, and an aramid nanofiber network reconstructed from 3D tomographic images. Because this contribution is concerned with graph machine learning benchmarks, all our materials are networks that have previously been shown to benefit from graph-based representations. In Section 3.4, we quantify the structural complexity of these materials.

### 3.1 POLYMER NETWORK

Polymer networks are formed by crosslinking polymer chains at multifunctional junctions and are widely used for tissue engineering, implantable devices and drug delivery (Li & Mooney (2016); Nonoyama & Gong (2021); Yuk et al. (2022)). A central challenge is understanding how these networks fracture, since performance is often limited by when and where failure occurs. The underlying network topology (the arrangement of chains and junctions) plays a critical role in fracture behavior (Dobrynin et al. (2023)). In molecular dynamics simulations, both generating these networks and simulating their fracture require very large systems (tens of thousands of particles) to capture macroscopic behavior, demanding thousands of CPU hours on high-performance computing resources.

We introduce a structured dataset derived from coarse-grained molecular dynamics (MD) simulations that frames a long-standing materials science problem as a machine learning benchmark. The MD simulation protocol follows established approaches (Barney et al. (2022)). Briefly, the simulations mimic experimental systems where linear polymer chains with reactive end groups crosslink with tetrafunctional junctions, and the resulting network structures are represented as graphs: junctions are modeled as nodes and polymer chains as edges. Graph representations naturally encode topological defects such as self-loops, parallel edges, and higher-order cycles. Prior work has shown that polymer chains with fewer local topological defects, higher-than-average geodesic edge betweenness centrality, and stronger alignment with the loading direction are more likely to break under uniaxial tensile deformation (Zhang & Riggleman (2024)). These insights highlight opportunities for the ML community to apply graph models to predict fracture behavior directly from network topology.

We provide ensemble datasets from 10 combinations of polymer mole fraction ($\phi$) and chain length ($N$), which produce networks with varying topology and defect concentrations. Because fracture is not fully deterministic and thermal fluctuations influence which chains break, we performed isoconfigurational ensemble simulations, where the same network structure was fractured multiple times with randomized initial particle velocities. This approach yields per-chain breakage probabilities ($P_{break}$) and sequences of break events. The benchmark dataset supports two core tasks: 1) Link prediction: recovering missing connections in partially observed networks, providing a pathway to accelerate generation of large-scale polymer systems. 2) Fracture prediction: predicting where

and in what order polymer chains break under uniaxial deformation, directly linking microscopic network topology to macroscopic failure behaviors.

## 3.2 NANOWIRE NETWORKS

Nanowire networks show unique combinations of transparency, flexibility, and conductivity, making them ideal for flexible electronics and electrical shields for aerospace applications (Wu et al. (2024); Tan et al. (2020)). Surrogate random stick models have been used by theorists to study nanowire networks (Jagota & Scheinfeld (2020)). However, these were recently shown to differ from the real complex structures (Wu et al. (2024)), which was shown to have an impact on their properties. While the infeasibility of testing all possible real networks motivates their ML-driven study, these materials are distinctly challenging to collect data for. Specialized laboratory expertise and equipment for material synthesis, including a purpose-built novel flow system that allows the optimal air-solvent mixture required for film synthesis, is required. Property measurement for the resultant network then requires running four-point probe experiments specifically designed for assessment of conductive film resistances. Advanced scanning electron microscopy is then required for resolving nanowire network structure (costing $100k-$1mn).

In order to open the possibility for data-scare model development, we present this dataset of 33 graphs across four sets of scanning electron microscopy images of silver nanowires. Each set corresponds to a sample with a particular electrical sheet resistance that was measured experimentally, using the well-established four-point probe technique, with experimental measurement error less than 10 %. Each image has a graph associated with it, which was extracted with *StructuralGT*, as depicted in the top left quadrant of Figure 1. For this dataset, we test the ability for graph learning to predict the electrical resistance of the networks from the extracted graphs.

The challenge in developing ML models for this dataset thus far comes from the combined difficulties of the above mentioned experimental synthesis, imaging, and characterization, leading to data-scarcity.

## 3.3 ARAMID NANOFIBERS

Synthesized from well-known Kevlar, aramid nanofiber networks show enhanced thermal and mechanical properties (Yang et al. (2011)), and have since shown application in membranes (Wang et al. (2022)), structural batteries (Wang et al. (2020)), and protective coatings (Wang et al. (2025b)). Aramid nanofiber networks take even longer to synthesize than nanowire networks ($\sim$ 10 days). To confidently predict their complex, multidirectional behavior from their structure, accurate 3D imaging is required. Our nanofiber graph is the first public example from a 3D tomographic image, processed by *StructuralGT* and yielding a graph of over 330,000 nodes. For this dataset, we test a series of models' abilities to predict the presence of links when the graph is incomplete.

## 3.4 DATASET COMPLEXITY

While our material graphs come from a diverse range of experimental and computational sources, they share a common feature: structural complexity. Although quantitative complexity measures remain an open question in materials science, here we adopt the general heuristic of a mix of order and disorder/defects as a hallmark of complexity, commensurate with information theory and thermodynamics (Mao & Kotov (2024)). Leveraging a wide range of graph theory metrics, we contrast the highly ordered and disordered structures with our own structures in Fig. 2. The clustering and square clustering distributions for all networks are shown in Fig.2 a-l. We also calculate generalized fractal dimensions (GFD) and node-based multifractal analysis (NMFA) for topological complexity Xiao et al. (2021). In Fig.2o we show standard parameters quantifying local environments, such as degree, clustering, square clustering, and degree assortativity. We also include graph Ollivier-Ricci curvature (ORC), which is a crucial feature in identifying the community structure of complex networks (Sia et al. (2019)), and asymmetry from the analyses in (Xiao et al. (2021)). Separate from graph theory, nematic order parameter (Wu et al. (2024)) is used to measure the correlation between orientations of geometric entities, and increases with increasing orientational order. Full definitions are given in Appendix C.

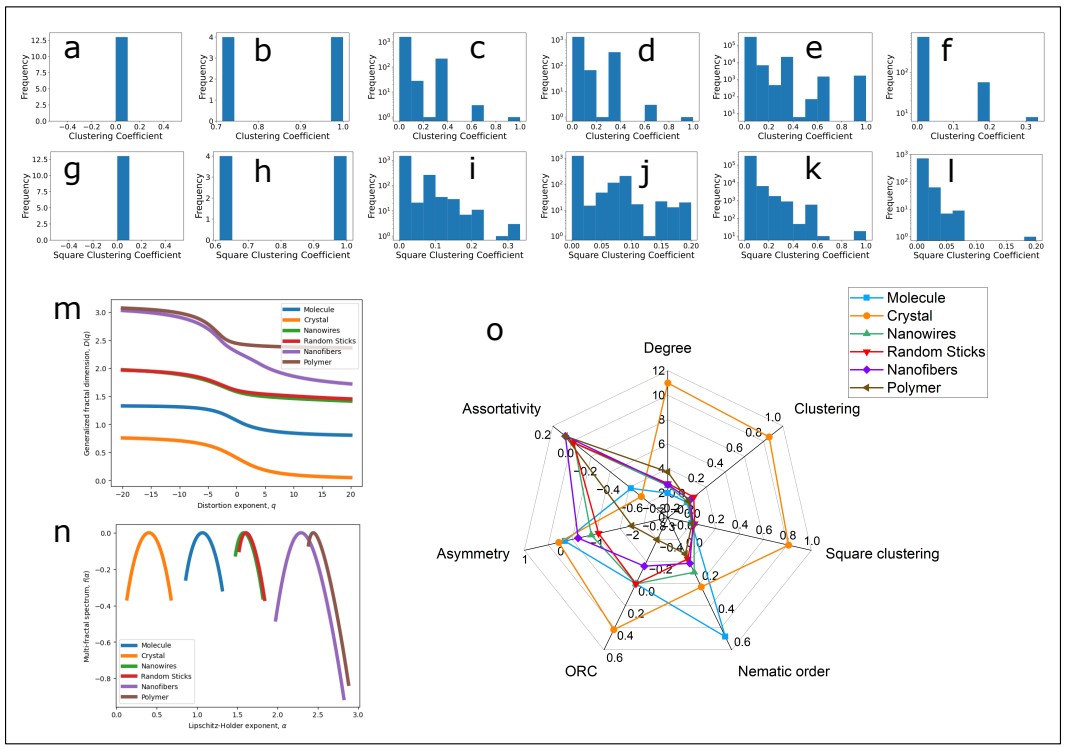

Figure 2: Clustering coefficient distributions for graphs from a variety of material. (a) Molecular graph of aspirin, from PubChem; (b) Unit-cell for zincblende, from CHILI (Friis-Jensen et al. (2024)); (c) A graph from an experimental image of a nanowire network; (d) A random stick models created for this work; (e) A graph from a 3D tomographic reconstruction of aramid nanofibers; (f) A graph from the molecular simulations of polymer networks. (g-l) is the same but for square clustering coefficients. (m) Generalized fractal dimension of each graph; (n) Multi-fractal spectrum of each graph; (o) a radar plot depicting the values of some graph theory metrics and order parameters or each of the graphs (average degree, average clustering coefficient, average square clustering coefficient, nematic order parameter, average Ollivier-Ricci curvature (ORC), asymmetry, degree assortativity. The nanowire and random stick values have been normalized w.r.t. their ordered disordered limiting values for 2D structures (see Appendix C).

First, because of stoichiometric constraints not present in most complex materials, the molecular graph for aspirin has (square) clustering coefficient of $0$ for all atoms (Figure 2 a,g). Due to its simplicity and small size, it has near-zero asymmetry, and high nematic order (Figure 2o). For the crystal unit cell, high order leads to (square) clustering coefficients taking only one of two values (Figure 2 b,h). High density of connections leads to high clustering values, degree and ORC, shown in Figure 2o. Additionally, it can be shown that there are only 19 unique orientations of the bonds, which results in the high nematic order parameter. As with molecular graph, the asymmetry is near-zero, suggesting no dominant structure. The exponent $\alpha_0$ from NMFA - the $\alpha$ value at maximum $f(\alpha)$ - is crucial to quantify the complexity of network structure. The higher $\alpha_0$ indicates the higher degree of complexity in network structure (Xiao et al. (2021)). From Figure 2 n, both crystalline unit cells and small molecule graphs - common targets of graph learning studies - manifest low exponent $\alpha_0$, therefore do not exhibit complex structure.

Meanwhile, the NMFA spectra show significantly higher $\alpha_0$ in nanofibers and polymer datasets, demonstrating higher degree of complexity in their networks (Fig.2 n). The negative asymmetry implies thorn-like structures dominate in the network. To highlight how hidden geometric order may result in unique complex material properties relevant to their application, we can compare complex nanowire network results to results from the commonly used random stick surrogate model. Comparisons show that, despite having very similar clustering distributions, GFDs and NMFA (Fig.2 c,d,i,j), the real networks have significantly higher hidden nematic order (Figure 2o). This is because

Table 1: Statistics of datasets.

| Dataset | #Graph | #Node (per graph) | #Edge (per graph) | Diameter |
|---------|--------|-------------------|-------------------|----------|
| Nanofibers | 1 | 336968 | 901066 | 154 |
| Nanowire | 33 | 712 - 3899 | 1670 - 9162 | 50 - 128 |
| Polymer | 10 | 800 - 8000 | 1493 - 15642 | 12 - 44 |

Table 2: Evaluation of the Link Prediction on ComMat datasets. The metric used is AUC ↑. Best is bolded and runner-up is underlined.

| Model | Nanofibers | | | Nanowire | | | Polymer | | |
|-------|------|------|------|------|------|------|------|------|------|
| | 5% | 10% | 20% | 5% | 10% | 20% | 5% | 10% | 20% |
| GCN | 63.60±1.10 | 63.02±1.03 | 59.14±0.80 | 75.00±1.84 | 73.04±1.74 | 71.90±3.70 | 80.68±0.27 | 79.47±0.25 | 78.36 ± 1.11 |
| GAT | 56.40±0.55 | 55.23±0.78 | 53.10±0.65 | 66.80±1.30 | 67.52±1.65 | 64.46±2.71 | 76.18±0.69 | 77.04±0.51 | 77.10±0.44 |
| GraphSAGE | **77.19±0.44** | **74.95±0.17** | **68.54±0.92** | 66.13±1.86 | 62.70±1.34 | 56.64±1.99 | 78.36±1.34 | 78.14±0.39 | 77.41±0.48 |
| Centrality-GCN | 68.78 ± 2.93 | 65.63±2.36 | 61.56 ± 1.80 | **77.62 ± 1.18** | **74.45±1.93** | **72.13 ± 3.70** | 81.45 ± 0.16 | 79.24±1.16 | 77.89 ± 0.85 |
| Centrality-SAGE | 76.80 ± 0.44 | 74.84±0.46 | 67.53 ± 1.23 | 66.43 ± 2.35 | 63.85±1.90 | 57.30 ± 2.31 | **83.26 ± 2.88** | 80.48±2.63 | 76.97 ± 0.48 |
| EGNN | 73.02±11.9 | 73.44±10.6 | 67.98±12.9 | 62.73±11.7 | 55.10±8.33 | 55.61±9.73 | 72.68±0.71 | 75.43±1.12 | 75.50±0.28 |
| Equiformer | 76.14±10.3 | 63.51±11.5 | 61.52±10.1 | 59.54±1.28 | 53.92±1.80 | 54.63±1.58 | 82.91±0.09 | **82.57±0.48** | **80.24±0.94** |

random stick networks are generated from i.i.d. sampling of orientations and positions, while real nanowire networks experience crowding effects, resulting in alignment of neighboring edges, and hence correlations in the orientations. Because it is a geometric effect, the discrepancy is not detected by most topological analyses (Fig.2m-o). Although subtle, this structural difference has been shown to impart unexpected directional dependencies of properties (Wu et al. (2024)).

Thus, both graph theory and materials metrics confirm the presence of complexity in our proposed datasets. Additionally, the presence of defects in the polymer network is enumerated by a considerable number of nodes with non-zero clustering coefficients (Figure 2 f), which should be zero for all nodes in ideal networks (Zhang & Riggleman (2024)).

# 4 EXPERIMENT AND BENCHMARKING

In this section, we elaborate on the experimental setup for our proposed datasets, where the overview of statistics is provided in Table 1. We perform experiments on comprehensive standard graph learning models, evaluating their effectiveness and discussing potentials for future research.

## 4.1 EVALUATION TASKS

**Link prediction** Link prediction aims to predict missing links from incomplete graph, which is a fundamental task to evaluate model capability to capture the network structure. Link prediction is crucial for graphs derived from both experimental imaging and MD simulations because microscopy-based skeletonization often introduces missing or spurious edges, and generating networks via MD simulations can be computationally expensive. A learned link prediction model can denoise experimental graphs by distinguishing true physical connections from imaging artifacts, and can also approximate MD-generated connectivity patterns at a fraction of the computational cost. Robust link prediction performance enables more accurate, scalable, and physically meaningful graph representations for studying structure–property relationships in complex materials.

We conduct the link prediction as a binary classification task, predicting whether there exists a link between two nodes. For Nanowire and Polymer datasets, we use the largest graph for this task. For each graph, we retain a portion of total links for validating and testing, with the ratio of 5% / 10% / 20% sequentially. The models are trained on the subgraph constructed from the remaining links. The ratio of positive and sampled negative edges is set as 1:1. The performance is measured by the Area Under Curve (AUC).

**Resistance prediction** We formulate the property prediction task on the Nanowire dataset as a graph regression problem, which aims to predict the electrical resistance for each graph. For each set of Nanowire, we split the total number of graphs into 60% / 20% / 20% for training, validation,

Table 3: Evaluation of resistance prediction for Nanowire. Best is bolded and runner-up is underlined.

| Metrics | GCN | GAT | GraphSAGE | EGNN | Equiformer |
|---|---|---|---|---|---|
| MAE ↓ | 61.48 ± 0.38 | 60.98 ± 0.29 | 62.83 ± 1.68 | 59.89 ± 0.12 | **57.46 ± 2.76** |
| RE ↓ | 0.49 ± 0.01 | 0.49 ± 0.01 | 0.48 ± 0.01 | 0.51 ± 0.01 | **0.39 ± 0.07** |

Table 4: Evaluation of the link breaking probability for polymer. Best is bolded and runner-up is underlined.

| | GCN | GAT | GraphSAGE | EGNN | Equiformer |
|---|---|---|---|---|---|
| mMAE ↓ | 0.30 ± 0.03 | **0.29 ± 0.03** | 0.32 ± 0.05 | 0.31 ± 0.03 | 0.50 ± 0.01 |

and testing, respectively. We use the mean absolute error (MAE) and relative error (RE) to measure the deviation of predictions from ground truth.

**Fracture prediction** Predicting fracture in polymer networks is crucial for evaluating material performance across a wide range of applications. We design this task as link regression problem, where the model predicts the breaking probability $P_{break}$ for each target link. For each graph, the total links are partitioned into training (60%), validation (20%), and testing (20%) sets. To maintain the balance in ground-truth distribution, we sample a maximum of 100 links for each label. MAE is used to measure the error for each graph, and we report the mean MAE (mMAE) for the overall performance of all graphs.

## 4.2 GRAPH LEARNING MODELS

**Classic GNNs** Classic GNNs adopt the MP paradigm to learn from graph-structured data, where each node iteratively updates its feature representation by aggregating information from its neighbors. Among that, GCN Kipf (2016) aggregates information by operating convolution, which is approximately equivalent to mean pooling over the neighbor nodes. GAT Veličković et al. (2017) applies a self-attention mechanism to assign weights to each neighbor, thereby modeling their levels of importance. SAGE Hamilton et al. (2017) concatenates information from its neighbors with the target node representation during aggregation, ensuring that the node's original identity is preserved and plays a key role in its new embedding.

**Centrality-aware GNNs** We adopt centrality encoding Ying et al. (2021), which embeds node degree centrality together with features vector, to popular GNN methods such as GCN and Graph-SAGE, denoted as Centrality-GCN and Centrality-SAGE.

**Geometric GNNs** These methods leverage spatial information to learn equivariant graph representations under geometric transformations, such as translation, rotation, and reflection. E-GNN Satorras et al. (2021) employs equivariance by incorporating relative squared distances into the MP function and updating node coordinates based on relative position vectors. Equiformer Liao et al. (2023) leverages irreducible representations to integrate SE(3)-equivariant features into the network's channels, preserving geometric information without altering the graph structure.

## 4.3 RESULTS

**Link prediction** Results from Table.2 show that existing baselines can achieve promising performance on this task. Overall, centrality-aware GNNs can match or outperform their classic counterparts, suggesting that incorporating centrality could be beneficial and promising for future development. Geometric GNNs often show higher fluctuation performance while being less computationally efficient. This might indicate that the geometric equivariance properties are not as beneficial for this particular link prediction task, and these models may require careful hyperparameter tuning. Overall, the performance degrades as more of the graph structures are hidden, reflecting challenges in structural missing.

**Resistance prediction**  The results in Table 3 highlight two findings: first, all baselines obtain suboptimal performance on this task, as shown by the high MAE and RE values. It shows that the task remains challenging for current mainstream GNN models, and more effective methods should be investigated. Second, geometric GNNs significantly outperform classic counterparts on this task. This suggests the geometric information is essential for physics-related problems, where target properties could be intrinsically governed by the material's physical shape.

**Fracture prediction**  We also witness the underperformance of benchmark graph methods on this task, as shown by the high MAE values in Table.4. The classic GNNs tends to be more robust than geometric class, suggesting the local connectivity patterns could be more effective for predicting the breaking probability.

**Discussion**  Overall, there is no single model that dominates for all tasks. We find that benchmark models are unable to predict behavior that has previously been shown to be deterministic and predictable from complex material graph structure (Wu et al. (2024); Zhang & Riggleman (2024)). This may arise from a combination of complex material data scarcity and from the MP's inability to efficiently learn features uniquely crucial to predicting complex material performance. Specifically, betweenness centrality parameters and edge-flows (as calculated from networked linear transport methods) are known to be strong predictors of complex material performance. However, those long-range connectivity patterns are computationally expensive, limiting their utilization on large-scale graphs. Experimental results show that integrating simple, local centrality can yield performance gain, suggesting the potential for further centrality-aware advances. Future research could investigate more effective, efficient approaches to incorporate global indicators with local aggregation to enhance performance.

While we acknowledge the limited data for the resistance prediction task, such data scarcity is an inevitable reality in complex materials as well as frontier research. We position this benchmark as a rigorous challenge, urging for the development of models capable of learning on extremely sparse data. We also invite the community to explore broader utilization of this dataset. A possible scenario is to leverage it as an independent, zero-shot benchmark for assessing materials foundation models Miret & Krishnan (2025); Yan et al. (2025).

## 5   CONCLUSION

In this work, we present ComMat, the first open graph benchmark of complex materials to foster ML research for advanced materials science. ComMat consists of 3 complex materials datasets unified under graph representations from multi-modality sources, including previously unreported 3D nanonetwork graphs. We show how the complex material structures at this intersection exhibit structural features distinct from previous material datasets. By providing a range of material scales, properties, and predictive tasks, ComMat enables a thorough and challenging evaluation of machine learning models. Our benchmark experiments highlight a significant performance gap, showing that popular graph ML models currently struggle with the challenges posed by these complex materials. These limitations raise the urgent need for novel graph learning algorithms that can accelerate the design and discovery workflows for advanced materials.

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

APPENDIX

# A  3D NANOFIBER NETWORK

In Figure 3, we show a few snapshots of a very small subset of the nanofiber graph.

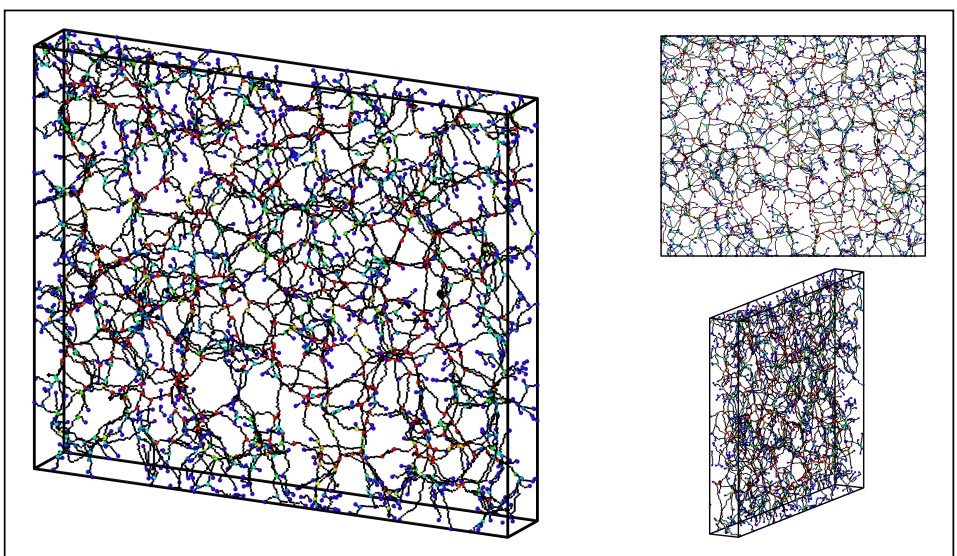

Figure 3: Snapshots of a subset of the Aramid Nanofiber graph used in this work. The nodes are colored by node betweenness centrality.

# B  MODEL SETTINGS

We provide the details of model parameters for each experiment tasks:

**Link tasks**   On link prediction and fracture prediction task, we employ a GNN encoder of 2 layers and the decoder as a multilayer perceptron (MLP) for all baselines. Each GNN layer is followed by a layer normalization, and the hidden dimension is set as 32. For GAT, we use 4 heads of attention. For GraphSAGE, we use mean pooling for neighborhood aggregation. For EquiformerV2, we set the maximum degree of spherical harmonics as 2. the dropout ratio is set as 0.5 for classic GNNs and 0.1 for geometric GNNs.

**Graph task**   For all baselines, we use a GNN encoder of 3 layers with the hidden dimension set as 32, followed by layer normalization. Finally, a graph mean pooling layer is applied to obtain a graph embeddings from all element nodes. The regressor is implemented as 2-layer MLP.

# C  GRAPH PARAMETER DEFINITIONS AND CALCULATION METHODS

## C.1  AVERAGE DEGREE AND AVERAGE CLUSTERING COEFFICIENT

Average degree and average clustering coefficeint, $\Delta$, were calculated using the **structural** module from *StructuralGT*, and follow the standard definitions given in the documentation:

$$\Delta = \frac{\sum_i \delta}{n},$$

$$\delta_i = \frac{2 * T_i}{k_i(k_i - 1)}$$

where $T_i$ is the number of connected triples (visually triangles) on node $i$ and $k_i$ is the degree of node $i$.

## C.2 GRAPH OLLIVIER-RICCI CURVATURE

Ollivier-Ricci Curvature Sia et al. (2019) measures the local geometric property in the network. An edge with positive curvature suggests it resides in a tight cluster, while a negative curvature edge tends to be a "bridge" between clusters of the network.

$$\kappa(x, y) = 1 - \frac{W_1(\mu_x, \mu_y)}{d(x, y)},$$

where $d(x, y)$ is the distance between two points $x$ and $y$, $W_1(\mu_x, \mu_y)$ is the 1-Wasserstein distance (or Earth Mover's distance) between two probability distributions.

## C.3 NEMATIC ORDER PARAMETER

The nematic order parameter was calculated using the **geometric** module of *StructuralGT*. The nematic order parameter $S$ is defined via the eigenvalues of the nematic tensor, $\mathbf{Q}$. The nematic tensor is defined as

$$\mathbf{Q} = \mathbf{M} - \frac{1}{3}\mathbf{I}$$

where $\mathbf{M}$ is computed from a sum of multiplications of the vectors, $\mathbf{m}^{(i)}$, describing particle orientations:

$$\mathbf{M}_{\alpha\beta} = \sum_{i=1}^{N} \mathbf{m}_{\alpha}^{(i)} \mathbf{m}_{\beta}^{(i)}.$$

The eigenvalues of $Q$ are $\frac{2}{3}S, -\frac{1}{3}S, -\frac{1}{3}S$. The closer a graph's value of $S$ is to 1, the greater the degree of corelation between edge orientations. When $S$ approaches 0, the graph's edges are uncorrelated.

For 2D networks, edges are confined to a plane, and so the lower limit for the nematic order parameter is 0.25, as shown in Wu et al. (2024). The values of nematic order parameter for the 2D networks in 2g (nanowires and random sticks) are normalized w.r.t. to this lower limit.

## D FURTHER RELATED WORK

**Biological materials**   Material datasets for non-ordered structures are highly prevalent throughout the biological communities. Owing to the small number of nodes in most of the molecular graphs from drug discovery studies ($\sim$100 nodes), such structures are well-suited to MP models (Batatia et al. (2022); Wang et al. (2025a)), or transformer architectures that treat molecular sequences as linguistic entities Owoyemi & Medzhidov (2023); Zheng et al. (2025).

Non-ordered materials that extend beyond this size include non-graph-based representations such as protein sequences. Most dataset-like studies involve applying new models to existing datasets and the most recent studies have acknowledged that the behavior of these structures, like complex materials, depends on more than just aggregations over local environments (Atkinson et al. (2025)). As such, there has been a recent push to extend beyond the transformer and message-passing architectures, especially towards diffusion models (Watson et al. (2023); Abramson et al. (2024)). Bottom-up development of diffusion models suited to the discreteness of protein sequences include Bayesian Flow Networks (Graves et al. (2023)). Extension of these models to graph-based representations is a critical missing piece in ML-driven complex material design and our dataset offers the best benchmark to test them.

## E FURTHER NETWORK ANALYSIS

Figure 4 shows degree distributions for each network. Unsuprisingly, the molecular and crystal graphs have simple degree distributions (Figure 4a,b). Random stick model and nanowire networks

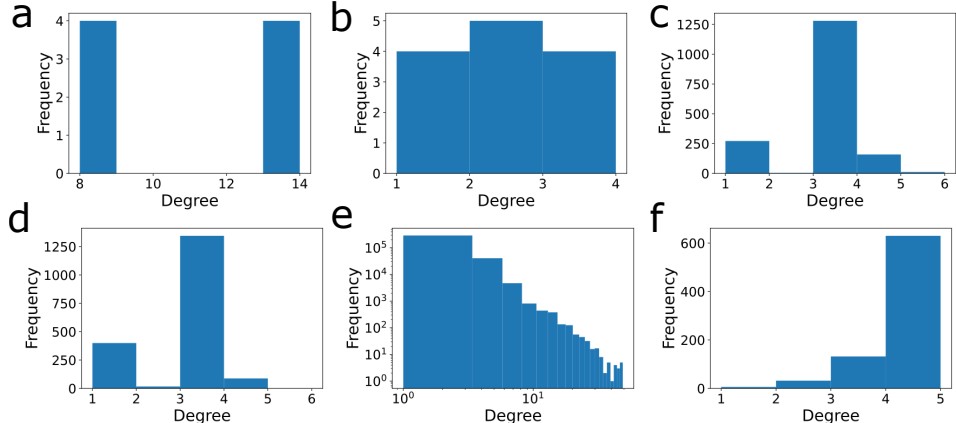

Figure 4: Degree distributions for graphs from a variety of material. (a) Unit-cell for zincblende, from CHILI (Friis-Jensen et al. (2024)); (b) Molecular graph of aspirin, from PubChem; (c) A graph from an experimental image of a nanowire network; (d) A random stick models created for this work; (e) A graph from a 3D tomographic reconstruction of aramid nanofibers; (f) A graph from the molecular simulations of polymer networks.

have very similar degree distributions (Figure 4c,d), which highlights the importance of geometric effects, as cpatured by the nematic order parameter in Figure 2o. Nanofiber network shows log-log distributions (Figure 4e), which is a commonly observed departure from the Poissonian distribution of random graphs, (and is often referred to scale-free (Barabási (2009))) While often attributed to the preferential attachment mechanism (Newman (2010)), geometric constraints mean that this doesn't apply here. The true source of this complex structure remains an open question, likely answerable by the ML community. Finally, polymer networks show defects, indicated by considerable number of nodes with degree less than four, which is the ideal network value (Zhang & Riggleman (2024)).

## F    FUTURE WORK

Acceleration of polymer network generation, which in current MD protocols requires costly simulations of chains and crosslinking junctions diffusing to form bonds, can be naturally framed as a link prediction task. While current models can capture some structural patterns, more fine-tuned or domain-informed approaches are needed to generate physically plausible networks with defect concentrations comparable to real materials. Limitations of this and similar directions include the data scarcity associated with the high computational cost of ground-truth data. However, this data scarcity also further motivates the development of models that generalize well on limited training data.

## G    ENVIRONMENT

For the polymer network dataset, molecular dynamics simulations were performed using the LAMMPS package (versions 09 Jan 2020 and 23 Jun 2022) Thompson et al. (2022) to generate model end-linked polymer networks and conduct fracture tests. Computational resources were provided by high-performance computing facilities under national and institutional programs.

Graph extraction and analysis for aramid nanofibers and silver nanowire network datasets were carried out on an Apple Mac mini 2020, with an Apple M1 CPU and 16 GB RAM, using a conda-forge distribution of StructuralGT 0.1.6 and Python 3.13.

The experiments for benchmark on graph ML tasks are conducted on a workstation including a 24-core CPU, 64GB RAM, and an RTX A6000 GPU. We implement the benchmarks using Python 3.10 and PyTorch 2.1 on Ubuntu-22.04.

