# OpenReview forum: "ComMat: Datasets and Benchmarks from complex materials for graph machine learning"
_ICLR.cc/2026/Conference — Submitted to ICLR 2026_

### Official Review · Reviewer_neFf · 2025-10-17

**Soundness:** 2
**Presentation:** 4
**Contribution:** 3
**Rating:** 6
**Confidence:** 4

**Summary:**

This paper presents ComMat, three computationally/experimentally obtained graph datasets of *complex materials*, defined here as order + disorder, including polymer network, nanowire networks, and aramid nanofibers. On the datasets, descriptive data analyses are performed, focusing on contrast with ordered and disordered materials families. Benchmark tests of graph neural network (GNN) models are conducted with link prediction, resistance prediction, and fracture prediction as the tasks.

**Strengths:**

- Constructing datasets and benchmarks for complex materials is timely and well-motivated, as graph representation and GNN models have demonstrated their capability in modeling simpler materials and gradually been adopted to complex materials.
- The paper is overall well-written and easy to follow, despite minor clarity issues (see Q4).
- The analyses of graph complexity using several metrics (Sec. 3.4) provide useful insights into understanding complex materials.

**Weaknesses:**

- The dataset size is limited, rendering the findings from benchmark tests thereon questionable (see Q1).
- Related to the previous point, the benchmark tasks could be better designed (see Q2).

**Questions:**

1.	The nanowire dataset contains 33 graphs, usually viewed as too few for GNN models. How much would lack of data (instead of models’ incapability) attribute to the models’ suboptimal performance? Also because of limited data size, the conclusion on benefit of geometric information (Line 450) is not backed up.
2.	Related to 1: Have the Authors considered other types of tasks (e.g., “local” or “node-level”), where the current data could be viewed as abundant?
3.	Some related studies on disordered materials and their graph-based models should be acknowledged: DOI: 10.1088/2632-2153/adc0e1; DOI: 10.1073/pnas.2322962121.
4.	Minor clarity issues
  - In Sec. 3.2, what are the nodes and edges of graphs? Besides, “33 graphs across four sets” and “experiments” are mentioned twice, which seems redundant.
  - In Sec. 4, when building GNN models, what are the node/edge features?
  - Sec 4.2 title “Baselines” is confusing, as there is no newly proposed model to be compared with these existing ones.

---

> ### Author Response · Authors · 2025-11-23
> **Author Response**
>
> We thank reviewer for valuable feedback. Regarding your concerns:
>
> > Q1: The nanowire dataset contains 33 graphs, usually viewed as too few for GNN models. How much would lack of data (instead of models’ incapability) attribute to the models’ suboptimal performance? Also because of limited data size, the conclusion on benefit of geometric information (Line 450) is not backed up.
>
> We share the challenges in obtaining the nanowire dataset and discuss its potential for ML research in general response.
> Due to the current limited size of dataset, we could not further evaluate the impact of lacking data on the model performance. However, research has shown incorporating physical inductive bias, such as geometric equivariance, demonstrates superior sample efficiency compared to standard models [1, 2].
>
> > Q2: Related to 1: Have the Authors considered other types of tasks (e.g., “local” or “node-level”), where the current data could be viewed as abundant?
>
> We did consider this but decided to instead focus on tasks that could be verified against practically relevant ground truths. For nanowire networks, this corresponds to experimental measurements of their conductivity; for polymer networks, this corresponds to molecular dynamic simulation of their fracture behaviour. For all materials, their microscopy-derived graph structure is relevant because of its impact on downstream modelling/machine learning tasks.
> As of now, there are no node classification/regression tasks that we think are relevant to the materials science community. It may indeed be possible to construct a task that predicts e.g. a node’s local environment, but this has little practical relevance regarding the material’s behaviour.
>
> > Q3: Some related studies on disordered materials and their graph-based models should be acknowledged: DOI: 10.1088/2632-2153/adc0e1; DOI: 10.1073/pnas.2322962121.
>
> Thank you for sharing. These are both very interesting examples of using local environments representations of disordered materials for graph learning. We have therefore added them to our collection of similar examples in Section 2.1.
>
> > Q4.1:  In Sec. 3.2, what are the nodes and edges of graphs?
>
> The edges are nanowire segments and the nodes are intersections between them, as depicted in the top left portion of Figure 1. We have now added a more explicit description in the caption.
>
> > Q4.2: Besides, “33 graphs across four sets” and “experiments” are mentioned twice, which seems redundant
>
> Thank you for pointing this out. We have now removed the second time this is mentioned.
>
> > Q4.3: In Sec. 4, when building GNN models, what are the node/edge features?
>
> The node features used are node coordinates.
>
> > Q4.3: Sec 4.2 title “Baselines” is confusing, as there is no newly proposed model to be compared with these existing ones.
>
> We thank reviewer for the feedback. We have updated the title to “Graph Learning Models” to avoid confusion.
>
> [1] Does equivariance matter at scale? Brehmer et al, TMLR 2025.
> [2] SE(3)-Transformers: 3D Roto-Translation Equivariant Attention Networks, Fuchs et al, NeurIPS 2020.

---

> > ### Comment · Reviewer_neFf · 2025-11-27
> >
> > The authors have addressed my concerns regarding clarity and writing. However, my major concern regarding dataset size remains unaddressed. Emphasizing *challenges* in data generation and *potentials* of ML for complex materials does not enhance the contribution of the current work; nor does this address the problematic evaluations of ML models due to lack of data. I will thus keep my rating.

---

### Official Review · Reviewer_jmek · 2025-10-25

**Soundness:** 1
**Presentation:** 2
**Contribution:** 2
**Rating:** 2
**Confidence:** 4

**Summary:**

The paper introduces a Complex Materials Benchmark (ComMat), with three datasets (experiment and simulation). Complex is (vaguely) defined as not falling into the category of crystalline periodic, nor amorphous materials, and thus not covered by current GNN approaches on either sides.

**Strengths:**

Introducing complex materials as a new graph learning application with interesting node/edge/graph regression/classification tasks and maybe even generative tasks is interesting and might be relevant for the materials AI community.

**Weaknesses:**

The definition of complex materials is not really clear; see questions. Overall, all introduced tasks are on networks on a coarse-grained level (no atom resolution). It is not fully clear if this is only a special case of what the authors mean by complex materials, and what the ordered aspects of this are, if the atom or even monomer resolution is given up. It makes sense that atoms in polymer networks have some aspects of order (along the backbone) and disorder (in between chains) in their local environment and radial distribution function. But on the CG level, this is unclear. Also, if only networks are studied here, this benchmark does not seem applicable to a broader definition of complex materials.

The meaning of the related work section seems unclear; see questions.

Generation time: The authors suggest the introduced data sets to be used as benchmark tasks. However, there seems to be a mismatch between the (rather small) amount of work that goes into generating the datasets and the expected amount of work (months to years of work plus tens of thousands of GPU hours) that might go into ML method development (assuming complex materials are relevant and interesting enough for the community to focus more on method development in this direction). Would it not make more sense to have larger datasets, more datasets, more complex in-distribution and out-of-distribution splits, more diverse complex materials classes (beyond networks), or just different types of polymers and nanowires/fibers if this should serve as a benchmark?

**Questions:**

"When materials are neither ordered nor disordered, but instead complex, ...". This seems not sufficient as a definition. Would a doped material or a material with point defects (structural order, occupational disorder) be complex according to this definition? Would a material with structural (1D/2D) defects be complex? "betweenness centrality" is also not clearly defined, even if it might be contained in Smart et al., but this definition seems highly important here so it should be well-defined and self-contained.

Related work: "we review existing datasets and benchmarks on materials science that do not fall into the complex material category, yet provides useful insight in our development of this contribution". It seems like you reviewed those datasets, but you did not discuss the useful insight they provide, nor the relation to your work.

Nanowire networks: "This dataset comprises 33 graphs": How are the graphs defined here? Why do you call those networks? Are they not just independent 1D wires? The same question applies to the aramid nanofibres. If those are not cross-linked like the polymer networks (?) then it sounds like the system consists of independent 1D fibers, and the graph definition remains unclear.

Link prediction: As the materials are (by definition) partially disordered and thus contain some degree of randomness: What is the upper limit that can be reached in a link prediction task, and what is the fundamental noise level? In the polymer dataset (but also the others), all models seem to reach rather similar performance, which might indicate that they are already close to the noise limit, and thus it cannot be expected that further model development will increase the AUC values. Learning curves (training/test AUC vs. training set size) would help to see potential effects of saturation.

Table 3: What are the units here? How is RE defined? What are the r2 values? What is the label distribution? How can a graph regression problem be solved with only 33 graphs and thus 33 labels?

Table 4: What is the definition of mMAE? mean over all links? Or graphs? What does "100 links for each label"? mean What are the labels (as you write this is a link regression problem)? What is the unit here, and what is the distribution of labels and errors?

---

> ### Author Response · Authors · 2025-11-23
> **Author Response**
>
> We appreciate reviewer feedback. Regarding your concerns:
>
> > W1: The definition of complex materials is not really clear; see questions. Overall, all introduced tasks are on networks on a coarse-grained level (no atom resolution). It is not fully clear if this is only a special case of what the authors mean by complex materials, and what the ordered aspects of this are, if the atom or even monomer resolution is given up.
>
> While the complexity definition we adopt here is scale-free (and thus could indeed be applied at the atomistic scale), in this work we are concerned with complexity that emerges at the CG level, because it is this level that most simultaneously (1) is controllable through variation of experimental conditions, (2) can be used to tune the properties of the material, (3) is challenging and so far very rarely quantified, and (4) the impact of CG-level complexity on material properties has not been studied with ML. CG complexity is also more complicated than its atomistic counterpart because the valency constraints inherent in atomic/molecular systems are relaxed. We agree that it was not clear which scale we were referring to (i.e. atomistic v.s. CG), so we have added clarifications to the main text.
>
> > It makes sense that atoms in polymer networks have some aspects of order (along the backbone) and disorder (in between chains) in their local environment and radial distribution function. But on the CG level, this is unclear.
>
> The coarse-grained (CG) model of polymer networks has been widely used in the polymer physics community to investigate structure–property relationships, ranging from equilibrium architecture to dynamical and mechanical behaviour, because of its efficiency and scalability. Improving classical polymer network theories, which often assume ideal or perfectly ordered network structures, by simulating networks with realistic distribution of defects has become a major research direction in recent years. Many efforts have led to notable progress in developing more accurate models of elasticity, fracture, and other mechanical properties.
>
> > Also, if only networks are studied here, this benchmark does not seem applicable to a broader definition of complex materials.
>
> While our definition of complex material may include non-network materials, we focus on materials that have previously been shown to benefit most from a graph-based representation - i.e. network materials, and thus leverage the graph ML research. We now have added this clarification to the main text. We would also like to add that – because we define complexity as the mix of order/disorder – popular material representations (unit cells, point clouds, sequences) are not applicable here and that graph-based representations have been the predominantly successful choice for complex material structure-property relationships.
>
> > Generation time: The authors suggest the introduced data sets to be used as benchmark tasks. However, there seems to be a mismatch between the (rather small) amount of work that goes into generating the datasets and the expected amount of work (months to years of work plus tens of thousands of GPU hours) that might go into ML method development (assuming complex materials are relevant and interesting enough for the community to focus more on method development in this direction). Would it not make more sense to have larger datasets, more datasets, more complex in-distribution and out-of-distribution splits, more diverse complex materials classes (beyond networks), or just different types of polymers and nanowires/fibers if this should serve as a benchmark?
>
> We absolutely agree that it would be great to have the kinds of larger datasets, complex splits, diversity, classes, etc. that the reviewer is referring to. However, this is not a realistic expectation, for reasons we outline in the general response to reviewers.

---

> > ### Author Response · Authors · 2025-11-23
> > **Author Response (Cont.)**
> >
> > > Q1: "When materials are neither ordered nor disordered, but instead complex, ...". This seems not sufficient as a definition. Would a doped material or a material with point defects (structural order, occupational disorder) be complex according to this definition? Would a material with structural (1D/2D) defects be complex?
> >
> > In this work, we chose a complexity definition general enough for motivating the current dataset. Namely, it highlights the structural features of complex materials (mix of order/disorder) that make their analysis with existing ML models inapplicable (reviewed in Related Work). For example, the lack of perfect order invalidates popular unit-cell representations. Additionally, the presence of non-random structural motifs distributed across large graphs makes methods such as message-passing—effective for molecular and other small graphs—insufficient for these materials.
> >
> > While there is a huge collection of complex materials, including those with dopants/defects that the reviewer is referring to, the purpose of this paper is to provide the first large, high-resolution dataset of complex materials for graph machine learning, with practically relevant benchmarks. Such benchmarks need well-established methods for their graph-based abstraction, as well as well-defined ground truths that can be used to construct ML tasks relevant to each material’s application, which significantly restrains the list of appropriate complex materials for this work. This is why we have chosen the collection of materials presented here.
> >
> > > Q2: "betweenness centrality" is also not clearly defined, even if it might be contained in Smart et al., but this definition seems highly important here so it should be well-defined and self-contained.
> >
> > Edge betweenness centrality, $EBC_{G}$, of an edge, $e$, is defined as
> > \begin{equation}
> >     EBC_{G}(e)=\frac{1}{2(N)(N-1)}\sum_{s,t \in \mathscr{N}}\frac{\sigma_{st}(e)}{\sigma_{st}}
> > \end{equation}
> >
> > We have now also defined betweenness centrality in the main text.
> >
> > > Q3: Related work: "we review existing datasets and benchmarks on materials science that do not fall into the complex material category, yet provides useful insight in our development of this contribution". It seems like you reviewed those datasets, but you did not discuss the useful insight they provide, nor the relation to your work.
> >
> > In preparing these benchmarks, we surveyed the current literature on graph learning material datasets. We learned that all previous studies use the message passing paradigm, which cannot easily learn centrality parameters, which have previously shown to impact complex material properties. Insight and relation to our work is the identification of this gap in the literature. We have now explained this in the Related Work section.
> >
> > > Q4: Nanowire networks: "This dataset comprises 33 graphs": How are the graphs defined here? Why do you call those networks? Are they not just independent 1D wires? The same question applies to the aramid nanofibres. If those are not cross-linked like the polymer networks (?) then it sounds like the system consists of independent 1D fibers, `and the graph definition remains unclear.
> >
> > Neither the nanowires nor the fibres are independent 1D components. In both cases, they form a percolating network, very similar to the polymer network (nanowires in 2D and nanofibres in 3D). The nanowire structure (as imaged by microscopy) and its graph-based representation are given in the top left portion of Figure 1. To clarify the graph definition here, we have added a more explicit description. Additionally, we have added a few snapshots of the 3D aramid nanofiber network in the Appendix.

---

> > > ### Author Response · Authors · 2025-11-24
> > > **Author Response (Cont.)**
> > >
> > > > Q4: Link prediction: As the materials are (by definition) partially disordered and thus contain some degree of randomness: What is the upper limit that can be reached in a link prediction task, and what is the fundamental noise level? In the polymer dataset (but also the others), all models seem to reach rather similar performance, which might indicate that they are already close to the noise limit, and thus it cannot be expected that further model development will increase the AUC values. Learning curves (training/test AUC vs. training set size) would help to see potential effects of saturation.
> > >
> > >
> > > > Q5: Table 3: What are the units here? How is RE defined? What are the r2 values? What is the label distribution? How can a graph regression problem be solved with only 33 graphs and thus 33 labels?
> > >
> > > Table 3 reports the MSE predicted and true electrical sheet resistance, which is an intensive analogue to conventional resistance, measured in ohms per unit square. The Relative Error (RE) is calculated as: $ RE = \frac{|y_{\text{true}} - y_{\text{predicted}}|}{y_{\text{true}}} $
> > >
> > > 33 nanowire graphs are extracted from 4 samples, each possesses a particular sheet resistance. The label distribution is as follows:
> > >
> > > | Resistance | # Graphs |
> > > |:--:|:--:|
> > > | 72 | 9 |
> > > | 95 | 5 |
> > > | 159 | 9 |
> > > | 244 | 10 |
> > >
> > > >Table 4: What is the definition of mMAE? mean over all links? Or graphs? What does "100 links for each label"? mean What are the labels (as you write this is a link regression problem)? What is the unit here, and what is the distribution of labels and errors?
> > >
> > > The labels are the link-wise breaking probabilities for each polymer graph $P_{break}$, calculated from 10 simulations per graph to yield discrete values in [0, 1] with a step size of 0.1. Since the label distribution is skewed toward lower probabilities (e.g., $0.1$) than higher ones (e.g., $0.9$), we mitigate imbalance by sampling a maximum of 100 links per $P_{break}$ value.
> > >
> > > MAE is used to measure the error of predicted and ground truth $P_{break}$ for all (testing) links within a single graph.  We then compute mMAE as the mean of MAE over all graphs, providing an overall evaluation.

---

> > > > ### Comment · Reviewer_jmek · 2025-11-25
> > > >
> > > > Complexity: If you focus on a particular scale, then this should be reflected in the title and abstract, etc. Your examples do not cover all aspects and scales of possible definitions of complex materials.
> > > >
> > > > Data scarcity: Your comment does not really answer my concern. If data generation is cheap (as you say in the manuscript. Of course, the equipment and the training to use the equipment is not cheap, but hundreds of labs worldwide have the same equipment), then you should generate a large dataset. This does not keep you from defining a benchmark which focuses on data-efficiency, but if you start from a small dataset, the tradeoff between accuracy and dataset size cannot be explored, which severely limits the usefulness of the dataset. You mention geometrical/physical inductive bias or transfer learning. Neither of these is applicable to your study, as the former is inherently problem-specific, so there cannot be a general, material-agnostic solution, and the latter is not possible with a very small dataset.
> > > >
> > > > Definition of complexity: "the purpose of this paper is to provide the first large, high-resolution dataset of complex materials for graph machine learning, with practically relevant benchmarks" --> This contradicts the previous answers. (a) You just argued that you are not providing a large dataset. (b) Your definition of complexity is very broad, but your examples are not. They are all network-type materials, so please rephrase your definition to fit your dataset. Otherwise, the title is too broad and misleading.
> > > >
> > > > Q4: The answer seems to be missing.
> > > >
> > > > Q5: My question for r2 values is not answered. Also, the label distribution is not continuous. Are you treating this as a regression or a four-class classification problem?
> > > >
> > > > Overall, I keep my rating. The definition of complexity used by the authors is too wide, given that all datasets are essentially networks. The datasets are small, making rigorous benchmarking hard.

---

> > > > > ### Author Response · Authors · 2025-12-02
> > > > > **Author Response**
> > > > >
> > > > > > Complexity: If you focus on a particular scale, then this should be reflected in the title and abstract, etc. Your examples do not cover all aspects and scales of possible definitions of complex materials.
> > > > >
> > > > > We selected our examples of complex materials based on two criteria: (1) they fit the definition of complexity in the paper, characterized by the coexistence of order and disorder in their structures, and (2) they have practical relevance, as outlined in the manuscript. We agree that many other classes of complex materials also meet this definition; in fact, this breadth highlights a central challenge we emphasize: the scarcity of high-quality datasets despite the widespread presence and importance of these materials.
> > > > >
> > > > > We have clarified our definition in both the abstract and the introduction. Furthermore, it is not feasible (and would not even be coherent) to collect “all aspects and scales of possible definitions of complex materials” in a single benchmark contribution. Our contribution instead aims to provide representative, well-motivated examples that illustrate the value and necessity of advancing ML methods in this domain.
> > > > >
> > > > > > Data scarcity: Your comment does not really answer my concern. If data generation is cheap (as you say in the manuscript.
> > > > >
> > > > > We absolutely did not say that data generation is cheap in the manuscript. In fact we have very explicitly added the opposite statements in our first update to the introduction:
> > > > > “One of the greatest challenges associated with building a complex material ML benchmark is the distinctly high cost associated with collecting complex material data. Graphs from experimental samples must first be synthesized, requiring specialized laboratory expertise, equipment, and materials. Then they must be imaged, requiring expensive and difficult to operate microscopy equipment. Finally, image-to-graph extraction requires novel image processing including binarization, skeletonization, and node/edge detection. Python APIs with ML library compatibilities have only just been released. In both experimental and computational cases, defining a practically relevant task associated with each material requires intimate knowledge of the material properties and industries that find them interesting. Combined with all the above necessitates highly interdisciplinary collaborations, which is an additional challenge.”
> > > > >
> > > > > > Of course, the equipment and the training to use the equipment is not cheap, but hundreds of labs worldwide have the same equipment), then you should generate a large dataset.
> > > > >
> > > > > The above claim about hundreds of labs worldwide having the same equipment is unsubstantiated. If you read our first general response to all reviewers, you will see that the flow system required for preparing the nanowire networks had to be built as a first of its kind. Similarly, the nanofiber dataset first requires preparation of nanofibers, which takes a post-doctoral level of expertise and weeks of their time, which is expensive. Second, imaging 3D structures of such materials with the currently employed method is also destructive - i.e. it destroys the sample in the process. **The 3D structure that we report here is the first time a 3D graph of a nanoscale network has been released.**
> > > > >
> > > > > > This does not keep you from defining a benchmark which focuses on data-efficiency, but if you start from a small dataset, the tradeoff between accuracy and dataset size cannot be explored, which severely limits the usefulness of the dataset.
> > > > >
> > > > > Waiting for ideal data conditions is not a viable strategy for scientific discovery, as obtaining data in frontier research is always challenging. For innovation, the ML researchers should embark on tackling constraints of specific application fields. As the ML community has successfully adapted to non-Euclidean data with graph learning, physical laws with physics-informed ML or noisy data with robust self-supervised learning, we believe that data scarcity associated with novel scientific fields like complex materials is the next constraint to overcome. Our datasets and benchmark lay the foundation to drive this evolution.
> > > > >
> > > > > You characterise our benchmark as small based solely on one dataset (nanowire) for one task (resistance prediction). However, our benchmark is a collection of 3 datasets of complex materials graphs, diverse in scales and tasks. The Nanofibers graph consists of more than 900,000 edges for the link prediction, which is comparable with previous benchmarks. The nanowire size is also sufficient for link prediction.

---

> > > > > > ### Author Response · Authors · 2025-12-02
> > > > > > **Author Response (Cont)**
> > > > > >
> > > > > > > You mention geometrical/physical inductive bias or transfer learning. Neither of these is applicable to your study, as the former is inherently problem-specific, so there cannot be a general, material-agnostic solution, and the latter is not possible with a very small dataset.
> > > > > >
> > > > > > Geometrical invariance/equivariance has been a mainstream research direction for molecular/material science graphs [1]. Transfer learning is the prominent approach to tackle limited data. Foundation models on materials, acquiring vast knowledge by pretraining on large datasets, can be leveraged to fine-tune or conduct few-shot learning on small datasets [2].
> > > > > >
> > > > > > [1] A Hitchhiker's Guide to Geometric GNNs for 3D Atomic Systems, Duval et al., Arxiv 2023.
> > > > > > [2] A Materials Foundation Model via Hybrid Invariant-Equivariant Architectures, Yan et al., Arxiv 2025.
> > > > > >
> > > > > > > Definition of complexity: "the purpose of this paper is to provide the first large, high-resolution dataset of complex materials for graph machine learning, with practically relevant benchmarks" --> This contradicts the previous answers. (a) You just argued that you are not providing a large dataset. (b) Your definition of complexity is very broad, but your examples are not. They are all network-type materials, so please rephrase your definition to fit your dataset. Otherwise, the title is too broad and misleading.
> > > > > >
> > > > > > (a) We would like to emphasize what it means for a dataset to be in different contexts. Conventional ML datasets associate “large” with a size of dataset much larger than what we are presenting here. However, our benchmark contains more samples than any existing complex material datasets (and, given the high data collection cost, is unlikely to be surpassed in the foreseeable future) and so is large in this context.
> > > > > >
> > > > > > (b) Our definition of complexity is consistent with materials science literature and so we will not change it. Our examples are chosen to include complex materials that have practical applications and well-defined ground truth tasks:
> > > > > > - Nanowire networks:
> > > > > >   - Flexible electronics for wearables
> > > > > >   - Transparent conductors for touchscreens
> > > > > >   - Protective coatings for electromagnetic shielding\
> > > > > >   - Strain sensors
> > > > > > - Aramid nanofibers
> > > > > >   - Structural batteries for aerospace applications
> > > > > >   - Mechanical shielding for protection
> > > > > >   - Membranes for water purification
> > > > > >   - Thermal management
> > > > > >   - Biomedical implants
> > > > > > - Polymer networks
> > > > > >   - Drug-delivery carrier
> > > > > >   - Contact lens
> > > > > >   - Tissue engineering
> > > > > >   - Rubber tires
> > > > > >   - Protective coatings

---

> > > > > > > ### Author Response · Authors · 2025-12-02
> > > > > > > **Author Response (Cont.)**
> > > > > > >
> > > > > > > > Q4: Link prediction: As the materials are (by definition) partially disordered and thus contain some degree of randomness: What is the upper limit that can be reached in a link prediction task, and what is the fundamental noise level? In the polymer dataset (but also the others), all models seem to reach rather similar performance, which might indicate that they are already close to the noise limit, and thus it cannot be expected that further model development will increase the AUC values. Learning curves (training/test AUC vs. training set size) would help to see potential effects of saturation.
> > > > > > >
> > > > > > > On nanowire and nanofibers datasets, the link prediction performance significantly varies among models. We conduct further experiments and provide learning curves on polymer datasets at https://anonymous.4open.science/r/ComMat-C8A9/.
> > > > > > >
> > > > > > > Overall, the train curves decrease and remain stable as the dataset size grows, reflecting typical learning behaviour. The test curve trends vary across models. GCN, Equiformer and GraphSage demonstrate improvement in generalisation when adding more data, while GAT, EGNN show slight improvement or plateau. The performance varies notably among different models (AUC from 76% to 80%). Thus, there is no evidence to claim that all current models have reached the dataset’s noise limit.
> > > > > > >
> > > > > > > Furthermore, the learning curves show the evaluated models' performance regarding data scale and cannot imply that more advanced model development will not improve the performance. In contrast, they indicate that architectural design and capabilities are crucial to better exploit the data and achieve improvement. While adding samples helps capable models (like Equiformer) improve, less expressive models may plateau regardless of data scale, similar to how a linear model cannot improve on complex, non-linear datasets simply by adding more samples.
> > > > > > >
> > > > > > > > Q5: My question for r2 values is not answered. Also, the label distribution is not continuous. Are you treating this as a regression or a four-class classification problem?
> > > > > > >
> > > > > > > $R^2$ is invalid for non-linear regression models, since the linear assumption for total variance does not hold for non-linear regression and can be misleading [1].
> > > > > > >
> > > > > > > The value of resistance (ohm per square) is continuous. As we have more data, the label distribution will vary and smoothen. We treat this task as regression for a general problem formulation.
> > > > > > >
> > > > > > > [1] "An evaluation of $R^2$ as an inadequate measure for nonlinear models in pharmacological and biochemical research: a Monte Carlo approach", Spiess et al., BMC pharmacology 10.1 (2010): 6.

---

### Official Review · Reviewer_go4Q · 2025-10-31

**Soundness:** 2
**Presentation:** 2
**Contribution:** 2
**Rating:** 4
**Confidence:** 2

**Summary:**

The present paper proposes a new benchmark called a complex materials benchmark (ComMat). ComMat focuses on complex materials where both ordered and disordered structures are present. Such materials not only show interesting characteristics but also challenge existing machine learning algorithms, because it is not trivial to mathematically represent the mixture of ordered and disordered structures. The authors propose this benchmark to motivate the community to develop algorithms for complex materials.

ComMat consists of three data sets. One is on polymer networks which are derived from molecular dynamics. Another is on nanowire networks where data are obtained by scanning electron microscope. The third one is on aramid nanofibers which are obtained by 3D tomography. The original data are converted into graph representations for down-stream tasks, and network complexity is analyzed by utilizing network scientific tools.

The authors define several prediction tasks and provided reference performance by baseline GNN-based methods. The results demonstrate that some of the tasks are far from being solved, suggesting that more sophisticated models have to be tailored.

**Strengths:**

- This paper generates data sets for benchmark.
- This paper defines several realistic tasks which are formulated as prediction tasks.
- The above strengths are essential for benchmarks.

**Weaknesses:**

The number of graphs in the Nanowire dataset is too small to formulate a graph regression problem. According to Table 1, the Nanowire dataset contains only 33 graphs, and the authors define a resistance prediction task on this dataset, which aims to predict the electrical resistance for each graph. Given such a small number of training examples, it is difficult to even evaluate the performance of predictive models.

Another concern is its relevance to real applications. While resistance prediction and fracture prediction are relevant to applications, it is not clear to me whether link prediction is relevant to real-world applications.

**Questions:**

- Are 33 graphs enough to formulate the resistance prediction task?
- It would be helpful if the authors could elaborate more on the importance of the link prediction task.

---

> ### Author Response · Authors · 2025-11-23
> **Author Response**
>
> > Q1: Are 33 graphs enough to formulate the resistance prediction task?
>
> We share the challenges in obtaining complex materials datasets, and discuss its potential for ML research in general response. Please have a check.
>
> > Q2: It would be helpful if the authors could elaborate more on the importance of the link prediction task.
>
> We thank the reviewer for this important question regarding the practical utility of link prediction in complex materials. The link prediction task is practically important for graphs derived from both experimental imaging and molecular dynamics (MD) simulations of complex materials.
> For experimental graphs, the accuracy of the extracted network can sometimes be limited by the resolution and noise characteristics of microscopy. Skeletonization (Fig 1 - main text) often introduces missing edges, spurious connections, or fragmented components, especially in dense or low-contrast samples. For example, Figure R1 from [1] shows the currently used manual process of removing spurious edges, required for refining graph-based representations of low-quality complex material images.
>
> A link prediction model that learns structural patterns from high-quality examples can help distinguish physically meaningful connections from imaging artifacts in lower-quality samples. Minimizing the noise in the resulting graph is critical for the study of complex materials, since several graph metrics (e.g., percolation thresholds, centrality) are highly sensitive to even small perturbations in the graph structure. Thus, the link prediction serves as a critical denoising and refinement step that makes experimental graph analyses more robust and physically interpretable.
>
> For computational graphs generated from MD simulations, link prediction can accelerate one of the most expensive parts of the workflow: simulating reactions in large systems until the network reaches a desired level of connectivity, disorder, or defect structure consistent with experimental observations. Pretrained link prediction models can approximate the connectivity patterns produced by MD at a fraction of the computational cost, enabling rapid generation of computational graphs with experimentally realistic disorder and connectivity patterns.
>
> In both settings, strong link prediction performance contributes directly to more accurate, scalable, and physically meaningful graph representations of complex materials. We have revised the 4.1 section to emphasize the practical importance of the link prediction task.
>
> [1] Structural Analysis of Nanoscale Network Materials Using Graph Theory, Vecchio et al, ACS nano 2021.

---

### Official Review · Reviewer_5ANq · 2025-10-31

**Soundness:** 2
**Presentation:** 2
**Contribution:** 2
**Rating:** 4
**Confidence:** 3

**Summary:**

This paper introduces ComMat, a novel benchmark suite of graph datasets derived from complex materials—systems that exhibit both order and disorder and therefore lie beyond conventional crystalline or amorphous material classifications. The authors unify data from diverse sources, including experimental microscopy, molecular dynamics simulations, and 3D tomographic reconstructions, into standardized graph representations to enable machine learning on these structurally intricate systems. ComMat comprises three datasets: Polymer Networks generated from 3D molecular simulations, Nanowire Networks extracted from 2D microscopy images, and Aramid Nanofibers reconstructed from 3D tomography. The benchmark defines multiple predictive tasks—link prediction, fracture prediction, and property (resistance) prediction—and evaluates a range of classical and geometric graph neural network (GNN) models. Experimental results reveal the limited performance of existing GNN architectures on these datasets, highlighting their difficulty in capturing the long-range connectivity and multi-scale structural dependencies characteristic of complex materials, and emphasizing the need for new graph learning approaches tailored to this emerging domain.

**Strengths:**

- ComMat is a new and useful resource for applying graph-based machine learning to study complex materials that have both order and disorder.

- It targets an important area that hasn’t been well represented in existing materials datasets.

- The authors combine data from different sources, such as experiments, simulations, and 3D imaging, into a single, easy-to-use format.

- This makes the dataset valuable for both materials scientists and machine learning researchers.

- They tested several well-known graph neural network (GNN) models, including GCN, GAT, GraphSAGE, EGNN, and Equiformer, on different prediction tasks.

- The source code is provided, and the dataset will be publicly released, making it easier for other researchers to use and extend.

**Weaknesses:**

- The core contribution lies in dataset creation and analysis, not algorithmic innovation. The benchmarking is fairly standard and doesn’t propose new learning architectures.
- The datasets vary drastically in size (e.g., one nanofiber graph with >300K nodes vs. 33 nanowire graphs), potentially limiting the generality of model comparisons.
- While the authors discuss the limitations of current GNNs for complex materials, they don’t experimentally test any adapted or domain-informed variants (e.g., centrality-aware message passing, hierarchical pooling).

**Questions:**

See the weaknesses

---

> ### Author Response · Authors · 2025-11-23
> **Author Response**
>
> We appreciate reviewer feedback. Regarding your concerns:
>
> > W1: The core contribution lies in dataset creation and analysis, not algorithmic innovation. The benchmarking is fairly standard and doesn’t propose new learning architectures.
>
> We submitted this paper to the Datasets and Benchmarks track because there is substantial effort required for extracting ML-compatible graphs for these kinds of data. Additionally, it requires laboratory expertise rarely found in the ML community. Hence, this contribution gives the community access to novel tasks that they would not otherwise have access to.
>
> Furthermore, we share our observations and insights to develop new, effective grahp ML methods for this kind of materials in our discussion, which are beneficial for future research. Proposing novel algorithms exceeds the scope of this track and should not hinder the contribution of our work.
>
> > W2: The datasets vary drastically in size (e.g., one nanofiber graph with >300K nodes vs. 33 nanowire graphs), potentially limiting the generality of model comparisons.
>
> The datasets vary regarding the materials, the data collection and processing. Specific tasks are designed for each dataset, and we train and evaluate each model separately for each dataset and task to ensure fair model comparisons.

---

> > ### Comment · Reviewer_5ANq · 2025-11-25
> >
> > I thank the authors for their rebuttal. However, I remain unconvinced regarding the selection of datasets, particularly given the extreme variance in size and structure across them. This raises concerns about the generality and fairness of the benchmarking results.
> >
> > Furthermore, my third concern remains unaddressed: although the authors discuss the limitations of existing GNNs for modeling complex material systems, they do not experimentally evaluate any adapted or domain-informed variants (e.g., centrality-aware message passing, hierarchical or multi-scale pooling). This gap limits the strength of their empirical claims.

---

> > > ### Author Response · Authors · 2025-12-02
> > > **Author Response**
> > >
> > > > I thank the authors for their rebuttal. However, I remain unconvinced regarding the selection of datasets, particularly given the extreme variance in size and structure across them. This raises concerns about the generality and fairness of the benchmarking results.
> > >
> > > We thank reviewers for your feedback. We first reemphasise the challenges in constructing these datasets, which are outlined in our general response (such as the flow system required for preparing the nanowire networks had to be built as a first of its kind). The collective requirements listed there are not available to all institutions and require researchers to form interdisciplinary teams that combine skills that must span different departments and research teams.
> > > Secondly, we want to address the point about generality. We are assessing each model on each dataset, the same way prior benchmark works assess each model on a collection of different datasets. We are not making any claims about model's ability to generalise to all datasets.
> > >
> > > We suppose that scale variety is an essential characteristic for a robust, impactful benchmark. Well-established graph benchmarks [1, 2] also provide datasets that span from small to large scale. This enables their utilisation for a wide range of research and tasks, from evaluating models that excel in low-resource environments to those capable of handling high-throughput, large-scale data.
> > >
> > > [1] Open Graph Benchmark: Datasets for Machine Learning on Graphs, Hu et al, Neurips 2020.
> > > [2] Temporal Graph Benchmark for Machine Learning on Temporal Graphs, Huang et al, Neurips 2023.
> > >
> > >
> > > > Furthermore, my third concern remains unaddressed: although the authors discuss the limitations of existing GNNs for modeling complex material systems, they do not experimentally evaluate any adapted or domain-informed variants (e.g., centrality-aware message passing, hierarchical or multi-scale pooling). This gap limits the strength of their empirical claims.
> > >
> > > We adapt Centrality Encoding from [1] to popular GNN methods, such as GCN and GraphSAGE, and evaluate their performance. Experiments show that incorporating centrality can achieve improvements or competitive performance, suggesting the potential for further centrality-aware development. More detailed results are provided in our revised version.
> > >
> > > | Dataset | GCN | Centrality-GCN | GraphSAGE | Centrality-SAGE |
> > > | :--- | :---: | :---: | :---: | :---: |
> > > | **Nanofibers** | 63.02 $\pm$ 1.03 | 65.63 $\pm$ 2.36 | 74.95 $\pm$ 0.17 | 74.84 $\pm$ 0.46 |
> > > | **Nanowire** | 73.04 $\pm$ 1.74 | 74.45 $\pm$ 1.93 | 62.70 $\pm$ 1.34 | 63.84 $\pm$ 1.90 |
> > > | **Polymer** | 79.47 $\pm$ 0.25 | 79.24 $\pm$ 1.16 | 78.14 $\pm$ 0.39 | 80.48 $\pm$ 2.63 |
> > >
> > > [1] Do Transformers Really Perform Bad for Graph Representation? Ying et al, Neurips 2021.

---

### Author Response · Authors · 2025-11-22
**General response to all reviewers**

We sincerely thank the reviewers for insightful and constructive feedback. We would like to use this reply to address common concerns shared by reviewers. We address other questions in individual replies.

> Regarding the concern of size of nanowire datasets (33 graphs), would that be sufficient for resistance prediction as graph regression task?

While we agree that our dataset is significantly smaller than conventional ML contributions, we would like to reemphasize data scarcity as an inherent feature of novel complex materials. The construction of  nanowire dataset requires:

- Specialized laboratory expertise and equipment for material synthesis, including a purpose-built novel flow system that allows the optimal air-solvent mixture required for film synthesis. Property measurement for the resultant network then requires running four-point probe experiments specifically designed for assessment of conductive film resistances. These require equipment and expertise different to those materials that the ML community is more familiar with (small molecules, antibodies, crystals, etc.).

- Advanced equipment is also required for imaging the structure. In this case, scanning electron microscopy is required for resolving nanowire network structure (costing ~\\$100k - \$1M). Using the equipment requires training far beyond that which is associated with conventional optical microscopy techniques.

- The image processing tools required for its graph-based extraction, which was only available recently as an open-source release of a Python API for integration with ML libraries from StructuralGT.

Hence the toolset required for experimental complex material graph generation is far different from the typical materials-to-data pipeline, whose experimental and/or computational configurations have been automated over decades for maximal data generation. Even if fully automated experimental workflows for the currently presented materials were to be developed, the constant push for new complex materials will inevitably involve novel experimental setups, which always include human labor bottlenecks, and subsequent data scarcity.

It is for these reasons of inevitable future data scarcity that this contribution is distinctly well-suited as a rigorous benchmark for complex material graphs. This forces the ML community to develop sample efficient methods, such as incorporating geometrical/physical inductive bias or transfer learning.

Data scarcity is a defining characteristic of frontier science. The ML community has successfully utilized small, high-quality datasets to advance methods in challenging, data-constrained domains. For example, the common graph classification benchmark MUTAG (chemical) has around 188 graphs [1], or the UCI Hepatitis dataset contains 155 instances [2].

The collective concern of data scarcity voiced by the reviewers has shown us that we did not make the above points clearly enough in the manuscript. Hence we have now added them to the manuscript.

[1] How Powerful are Graph Neural Networks? Xu et al, ICLR 2019.
[2] A systematic method for diagnosis of hepatitis disease using machine learning, Sachdeva et al, ISSE 2023.

---

### Author Response · Authors · 2025-12-03
**Revision Overview**

We thank reviewers for valuable feedback. We have refined our manuscript to explicitly clarify our contributions and address the concerns raised. Our major updates are summarized as follows:

- To address the comments on the dataset size, we have added an explicit paragraph detailing the data scarcity challenge in the Introduction. We expect that this will more clearly underscore the significance of our contribution to the ML community, as well as the reviewers. We also reemphasize it in the Conclusion section. We also make more explicit references to the fact that the 3D nanonetwork graph of 900,000+ edges is the first reported of its kind.

- To address the nanowire concerns specifically, we have added to section 3.2 a description of how the data-scarcity inherent in experimental complex material data collection requires the development of specialized ML models that this contribution will offer benchmarks for.

- We removed the “generation time” column in Table 1 because it does not appropriately reflect the substantial time and effort involved in constructing the experimental pipeline—from synthesizing samples to developing the Python APIs used to generate the graphs. Instead, we now describe the procedures and challenges associated with obtaining these complex materials datasets and emphasize that the scarcity of high-quality data remains a major bottleneck in the field.

- We have added a paragraph discussing the resistance prediction task on nanowire as a representative challenge that reflects the inherent data scarcity in complex materials research. We also encourage the community to extend the utility of this dataset, such as leveraging it as an independent, zero-shot benchmark for evaluating materials foundation model.

- We have included centrality-aware GNNs in our experiments. Results demonstrate that integrating centrality can be beneficial and promising for future research on complex material graphs. We also expanded our evaluation on the polymer dataset through more extensive hyperparameter tuning and updated link prediction results.

Collectively, we hope that these revisions address concerns regarding the significance of our contribution. By clearly highlighting that data collection for complex materials is far more challenging than domains typically associated with ML, we expect that it should be clear that the scale of data for which future models should be developed is also different.

---

### Meta-Review · Area_Chair_42wx · 2026-01-06

**Summary:**

This is a dataset paper and there have been many concerns by the reviewers.

**Reviewer Concerns:**

Most of the reviewers are not supportive to the paper due to limitations on the nature and usefulness of the dataset. These concerns are not addressed during rebuttals.

**Reviewer Scores:**

Most of the reviewers are not supportive to the paper.

---

### Decision · Program_Chairs · 2026-01-26

Reject